# Detection of Minor and Major Depression through Voice as a Biomarker Using Machine Learning

**DOI:** 10.3390/jcm10143046

**Published:** 2021-07-08

**Authors:** Daun Shin, Won Ik Cho, C. Hyung Keun Park, Sang Jin Rhee, Min Ji Kim, Hyunju Lee, Nam Soo Kim, Yong Min Ahn

**Affiliations:** 1Department of Psychiatry, Seoul National University College of Medicine, Seoul 03080, Korea; rune1018@gmail.com (D.S.); wandy04@naver.com (H.L.); 2Department of Neuropsychiatry, Seoul National University Hospital, Seoul 13620, Korea; hellojr1123@hanmail.net (S.J.R.); demonic89@naver.com (M.J.K.); 3Department of Electrical and Computer Engineering and INMC, Seoul National University College of Engineering, Seoul 08826, Korea; tsatsuki@nate.com (W.I.C.); nkim@snu.ac.kr (N.S.K.); 4Department of Psychiatry, Asan Medical Center, Seoul 05505, Korea; hkpark@gmail.com; 5Institute of Human Behavioral Medicine, Seoul National University Medical Research Center, Seoul 03087, Korea

**Keywords:** major depressive episode, minor depressive episode, dimensional approach, voice, machine learning

## Abstract

Both minor and major depression have high prevalence and are important causes of social burden worldwide; however, there is still no objective indicator to detect minor depression. This study aimed to examine if voice could be used as a biomarker to detect minor and major depression. Ninety-three subjects were classified into three groups: the not depressed group (*n* = 33), the minor depressive episode group (*n* = 26), and the major depressive episode group (*n* = 34), based on current depressive status as a dimension. Twenty-one voice features were extracted from semi-structured interview recordings. A three-group comparison was performed through analysis of variance. Seven voice indicators showed differences between the three groups, even after adjusting for age, BMI, and drugs taken for non-psychiatric disorders. Among the machine learning methods, the best performance was obtained using the multi-layer processing method, and an AUC of 65.9%, sensitivity of 65.6%, and specificity of 66.2% were shown. This study further revealed voice differences in depressive episodes and confirmed that not depressed groups and participants with minor and major depression could be accurately distinguished through machine learning. Although this study is limited by a small sample size, it is the first study on voice change in minor depression and suggests the possibility of detecting minor depression through voice.

## 1. Introduction

According to the World Health Organization (WHO), there were 322 million people suffering from depressive disorders worldwide as of 2017 and depression is the leading cause of non-fatal health loss, and the burden is increasing rapidly each year [1]. Based on the Diagnostic and Statistical Manual of Mental Disorders, Fifth Edition (DSM-V), a major depressive disorder can be diagnosed when five or more different depressive symptoms occur, including one or more of the following: (1) depressed mood or (2) loss of interest or pleasure, lasting more than two weeks or longer [2]. However, since 1992, the importance of minor depression has been gaining recognition. Minor depression was diagnosed as not meeting the full criteria of major depression, such as a short period of depression, not being satisfied with either depression or decreased interest, or having only four or fewer depressive symptoms [3]. The symptoms of minor depression may be less severe than those of major depression; however, the decrease in function, comorbid diseases, and outcomes are all similar to those of major depression [4]. Furthermore, minor depression contributes greatly to this economic and social burden [5,6]. In addition, the clinical significance of minor depressive disorder is indicated as a risk factor for major depressive disorder (MDD), and a system capable of early diagnosis is necessary to prevent deterioration of social function [7,8,9].

Currently, clinical depression diagnosis is mainly based on DSM-V and the International Classification of Diseases and Related Health Problems, 10th Edition (ICD-10) [2,10]. However, there are several diagnostic limitations based on the criteria outlined in these manuals. The primary criticism of the DSM-IV and ICD-10 criteria is that the diagnosis is based on the number and duration of symptoms, resulting in a non-dimensional view of depression [11]. Additionally, since the diagnosis is based on subjective symptoms, the diagnosis rate of MDD is inevitably lower among groups who tend to report symptoms on a reduced scale [12,13]. Furthermore, since there is no objective marker for diagnosis, the accuracy of the diagnosis varies depending on the practitioner who makes it. According to the results of a meta-analysis, the sensitivity of depression diagnosis by general practitioners was only 50.1%, while the specificity was 81.3% [14]. This indicates that depression in primary care, despite being well-detected, has numerous misclassifications. However, according to the guidelines, the first drug used for depression in bipolar disorder is not antidepressants, and in bipolar disorder, the use of antidepressants may cause hypomania, so it is important to diagnose depression early and accurately [15]. Therefore, it is necessary to obtain diagnostic assistance for depression, using objective indicators.

In psychiatric interviews, patients’ voices and speech are a standard by which clinicians judge patient symptoms. Typically, voice is an index that reflects the characteristics of the vocal cords, and speech is an index that includes speech speed and hesitation [16]. In interviews, it has been judged that patients have depression when their utterance decreases and the pauses in the middle of utterances increase [17,18]. With the proliferation of computer technology, voices can be quantified, and several studies have been conducted to investigate the association between voice and depression. In 1993, it was confirmed that the F0 variable, which reflects the dynamics and energy of the voice, is associated with depression [19]. Subsequently, several voice indicators, such as vocal jitter, glottal flow spectrum, and mel-frequency cepstrum coefficients (MFCCs), were found to be associated with the severity of depression [20,21,22]. Based on these studies, voice was proposed as a biomarker for depression [23]. In addition, with the increasing use of artificial intelligence in the medical field, research has made it possible to predict depression using artificial intelligence, based on voice differences [24,25,26]. However, in most of that research, voice was measured through certain tasks rather than psychiatric interviews. In this respect, there is a limitation that the subject’s natural language was not sufficiently reflected. Furthermore, in previous studies, corrections were not made for other conditions that affect the voice, such as taking antipsychotics. In addition, studies investigating voice changes in relation to minor depression have been insufficient [19,21,23,27,28].

To overcome these limitations, this study aimed to differentiate groups based on depressive episodes as a dimensional approach and identify vocal differences according to the state of the depressive episode. Voice features were extracted from semi-structured interview recordings. Based on these voice differences, our secondary aim was to predict the subject’s degree of depression using vocal values through machine learning. Therefore, in this study, we attempted to confirm changes in voice in minor depression, by addressing the limitations of previous studies related to voice in depression; to the best of our knowledge, this is the first research of this type. We hypothesized that the voice biomarkers would be capable of differentiating groups by depression severity.

## 2. Materials and Methods

### 2.1. Participants and Study Design

Subjects were recruited from the patient population who visited the outpatient clinic of Seoul National University Hospital for depressive symptoms. Participants’ ages ranged from 19 to 65 years. The control group was recruited through postings and online advertisements. The inclusion criteria comprised subjects who were able to read and understand the questionnaire independently. Subjects were excluded when a participant’s voice could not be secured due to neurosurgery, a history of substance abuse, or depressive symptoms caused by organic causes, such as epilepsy. Further exclusion criteria were a history of brain surgery or head trauma, an estimated IQ of less than 70, and a dementia diagnosis. All participants completed a written consent form based on the Helsinki Declaration during their first visit. The research procedure was approved by the Institutional Review Board of Seoul National University Hospital (1812-081-995). 

Subjects’ voices were recorded at the interview site during the Mini-International Neuropsychiatric Interview (MINI). A structured interview recording file with an evaluator of between 30 and 50 min was obtained for each subject. From the file, the subject’s time of utterance was recorded while the voice portion was extracted. Based on this technique, subjects’ voice files were obtained with an average duration of 1083 s.

Participants without subjective depression or any current depressive episode as indicated by the MINI were placed in the not depressed group (ND). Participants with subjective depression, but whose current depressive symptoms were not sufficient for major depressive episodes based on the MINI, were classified as belonging to the minor depressive episode group (mDE). Participants who reported subjective depression and confirmed that it is a current major depressive episode through the MINI were classified as belonging to the major depressive episode group (MDE).

### 2.2. Demographics and Antipsychotics

Clinical demographic information was collected, which included sex, age, socio-economic status (SES), psychiatric treatment eligibility, psychiatric drug use, and participants’ height and weight. When taking any type of medication except for antipsychotics, mood stabilizers, antidepressants, and benzodiazepines prescribed by psychiatrists, they were classified as taking ‘other medication’. BMI was calculated from the collected height and weight. Since antipsychotic drugs can affect the voice through extrapyramidal side effects, the dose was checked in this study and compared for each group [29]. To correct the cumulative effect of antipsychotics, the doses of antipsychotics being taken were converted into their equivalent based on the daily drug dose (DDD), which were then summed [30,31].

### 2.3. Questionnaires

All subjects were evaluated for depression using MINI version 7.0.2. The MINI is a structured interview that can accurately diagnose depression based on DSM-V diagnostic criteria [32,33]. Thirty-three subjects in the ND did not present subjective depression. It was further confirmed via the MINI that a major depressive episode was not currently applicable. 

The Hamilton Depression Rating Scale (HDRS) was used to evaluate the objective depression severity among the subjects. The HDRS, which is comprised of 17 items relating to depression severity, is rated using a 5-point Likert scale ranging from 0 (*not present*) to 4 (*severe*). A score of 17 or higher indicates moderate depression, while 24 or higher indicates severe depression [34,35,36]. 

The Patient Health Questionnaire-9 (PHQ-9) was used to evaluate the participants’ subjective depression. The PHQ-9 was developed as a screening scale for depression and comprises nine items that are rated using a 4-point Likert scale ranging from 0 (*not at all*) to 3 (*nearly every day*). Scores of 10 points or higher indicate moderate to severe depression [37,38,39].

Anxiety was evaluated using the Beck Anxiety Inventory (BAI) [40]. The BAI comprises 21 items that are rated using a 4-point Likert scale ranging from 0 (*not at all*) to 3 (*severely*). A score of 10 points or higher indicates mild anxiety, while 19 or higher indicates moderate anxiety [41]. Based on the findings of a meta-analysis conducted in 2016, pathological anxiety was suggested as an evaluation for scores from 16 to 20 points or higher [42]. 

In addition, previous research showed that impulsiveness was found to be associated with depression and anxiety in men [43]. Impulsivity was thus evaluated using the Barratt Impulsiveness Scale (BIS). The BIS was developed in 1959 to evaluate impulsive personality traits, and the 11th version of the scale (BIS-11) is currently the most widely used [44,45,46]. The BIS-11 consists of 30 questions that are rated using a 5-point Likert scale ranging from 1 (*never*) to 4 (*always*).

### 2.4. Voice Feature Extraction

Voice features were extracted from four aspects, namely glottal, tempo-spectral, formant, and other physical features. All features were primarily obtained within each utterance and subsequently averaged over the entire time interval. Glottal features comprise information on how the sound is articulated at the vocal cords, and are obtained by parameterizing each numeric after drawing a waveform [21,47]. The glottal closure instance (GCI) was calculated first, and subsequently calculated in various differentiations through iterative adaptive inverse filtering. Next, GCI and differentiation forms were integrated to estimate the glottal waveform. Since GCI should have a low value at this point, larger waveforms were smoothed. Three parameters, namely the opening phase (OP), closing phase (CP), and closed phase (C), were then extracted.

Tempo-spectral features are acoustic features mainly used in music information retrieval (MIR), which are extracted via an audio processing toolkit called “Librosa” [48]. This comprises the temporal feature, which refers to the time or length of the interval that participants continue an utterance, as well as the tempo, which considers the periodicity of the onset. Additionally, averaged spectral centroid, spectral bandwidth, roll-off frequency, and root mean square energy were used as spectral features.

Formant features refers to the information about formants that are conventionally used in phonetics, which are obtained through linear prediction coefficients or LPCs [49]. The formant represents the resonance of the vocal tract and can be understood as the local maximum of the spectrum. Thus, several principal components were calculated and extracted from them. The first to third formants were exploited and their corresponding bandwidths were obtained.

For other physical attributes, the mean and variance of pitch and magnitude, zero-crossing rate (ZCR) [50], and voice portions were utilized. The ZCR indicated how intensely the voice was uttered, and the voice portions indicated how frequently they appeared. After calculating the average of the ZCR for a particular utterance, frames with ZCRs below the average were defined as silent.

### 2.5. Statistical Analysis

Categorical variables among demographic and clinical features were compared and analyzed using the chi-square method, and a post-hoc test was performed using Fisher’s exact test. In the case of continuous variables, three groups were compared using the Kruskal–Wallis H test because data were not normally distributed, while the Mann–Whitney U method was used as a post-hoc analysis. However, in this study, several features of voice and speech were extracted and multiple comparisons were made. Therefore, in order to prevent type 1 error, the post-hoc test was performed once more with the Benjamini–Hochberg test method. For voice features, the normality test was not significant, and the N number was not sufficient. When comparing voice characteristics, it is necessary to include several covariates of demographics and clinical features in the analysis. Thus, among the values with skewness or kurtosis values of 2 or more and −2 or less, normality was corrected by performing log function processing when skewness was positive, and square processing when negative [51]. Clinical variables were not transformed, because then the meaning of cutoff and the statistical influence as a covariate would be altered. Subsequently, a three-group comparison was performed via ANOVA and the *p* value was corrected using ANCOVA for age, BMI, and use of other drugs, which were different between the three groups. Analyses were conducted using IBM SPSS Statistics for Windows, Version 25.0 (SPSS Inc., Chicago, IL, USA).

To date, machine learning approaches to detect depressed speech have included logistic regression (LR), Gaussian Naive Bayes (GNB), support vector machine (SVM), and multilayer perceptron (MLP) [52,53]. Therefore, in this study, after applying all four methods, the accuracy was compared. The input data consisted of 93 cases including the ND. Of these, 70% and 80% were used as training data, and the remaining 30% and 20% were used as prediction data for two scenarios, respectively. In principle, the model should be constructed with only the given training data. However, with a machine learning approach, it can be difficult to represent the feature space with the lack of an adequate sample size, especially when using MLP. Therefore, in this study, a small amount of noise was added to each item of the sample vector to reinforce the data, which were then utilized in the experiment; this model was labeled “augmented.” Meanwhile, the model using seven voice features related to the severity of the episode was labeled “selected”. Furthermore, LR, GNB, and SVM were implemented via a Python package called Scikit-learn, while MLP was implemented via Keras [54,55]. 

## 3. Results

### 3.1. Comparison of Demographics and Clinical Characteristics According to Depressive Episodes

A total of 93 subjects were recruited from 10 January 2019 to 30 April 2020. The 60 subjects presenting with depression as per the MINI results were further classified into groups of 34 subjects corresponding to major depressive episodes (MDE) and 26 subjects corresponding to minor depressive episodes (mDE). 

Females comprised 70–79% of participants, and age and SES showed no statistical difference. The mDE group utilized more medicinal drugs. The MDE group participants had the highest BMI. Additionally, the MDE group had the most participants taking antipsychotic medications, but the dosage was not statistically significant. 

Although the rate of diagnosis evaluated through MINI was different for each group, the analysis was conducted based on the criteria that satisfied the current depressive episode, regardless of the diagnosis. The ND group also included subjects with psychiatric diagnoses. However, at the time of recruitment, these subjects did not have psychiatric symptoms, but were diagnosed in MINI due to symptoms such as depression, anxiety, and mania that existed in the past. (Table 1)

### 3.2. Clinical Characteristics

The severity of objective and subjective depression and anxiety tended to increase according to the severity of the depressive episode. However, there was no statistically significant difference in impulsivity among the three groups. The difference between the MDE and mDE groups for anxiety was not statistically significant. (Table 2)

### 3.3. Voice Features

Findings based on the characteristics of the 21 extracted voice features revealed eight features showing differences in each group. The voice features that showed differences between the normal and mDE groups were spectral centroid, spectral roll-off, sq mean pitch, standard deviation pitch, mean magnitude, ZCR, and voice portion. In the mDE and MDE groups, there was only one statistically significant different voice feature: standard deviation pitch. Voice features did not show a tendency to change with increasing severity of the episodes.

After adjusting for age, BMI, and medicine usage, a total of seven voice features showed statistical significance: spectral centroid (*p* = 0.008), spectral roll-off (*p* = 0.012), formant BW2 (*p* = 0.040), sq mean pitch (*p* = 0.027), standard deviation pitch (*p* = 0.020), ZCR (*p* < 0.001), and voice portion (*p* = 0.020). Additionally, the Jonckheere–Terpstra test was used to confirm whether there was a sequence for each group. All seven variables increased or decreased in the order of ND, MDE, and mDE, respectively (see Table 3 and Figure 1).

### 3.4. Prediction of Depressive Episode through Machine Learning

Basically, all 21 voice characteristics were used to construct the model. Meanwhile, in the ‘selected model’, seven negative features showing differences in each group through the Benjamini–Hochberg test (Table 3) were used. For the cases with augmented input vectors, the number of the training examples reached 100 times that of the original cases. The results were recorded via mean (maximum) of the best test set accuracies for the trials, namely three and five times for the 7:3 and 8:2 splits, respectively. For specification of the MLPs, the hidden layers of sizes 128 and 64 were used, and dropout was not applied. For evaluation, we adopted accuracy, area under the curve (AUC) with confidence interval 95%, precision, recall, and F1 score, as used in conventional machine learning analysis.

In general, the MLP indicated the best performance. Additionally, more training data guaranteed better performance for the MLP, reaching the highest mean accuracy for the 8:2 cases. The best result for episode severity was obtained with non-selected features and augmented data, which used MLP; the precision average was 65.6 while the recall average was 66.2. After calculating the area under the curve (AUC) through MLP, the findings with regard to the 7:3 training set showed that the AUC was 0.79 and 0.58 for minor and major episodes, respectively. In the 8:2 training set, the predicted value was 0.69 and 0.67 for minor and major episodes, respectively. (Table 4)

Additionally, LR and GNB showed an AUC of 58.8–64.7. However, sensitivity and specificity were 41.6~57.2, which did not indicate any better function than MLP. Furthermore, in the LR and GNB models, as the training set increased, no tendency to improve performance was observed. (Figure 2)

## 4. Discussion

In this study, structured interviews were used to examine the depressive episodes of participants, as well as to record their voices. Extracts of the subjects’ voices were subsequently examined and analyzed with the aim of investigating whether depression severity could be determined by voice characteristics.

In this study, participants’ voices were extracted and analyzed as 21 features. Among the 21 various indicators, several factors were included, such as the average pitch (reflecting the characteristics of the voice) and the ratio of the actual utterance to the utterance time (reflecting speech delay). The spectral centroid refers to the center of the voice spectrum and represents the degree of the speaker’s voice [56]. The spectral roll-off is the frequency below a specified percentage of the total spectral energy; clinically, the higher the amount of utterance of the treble, the larger the spectral roll-off [57]. Formant is defined as a broad peak or local maximum in the spectrum of spoken speech. The formant BW2 value is the second peak value and is a characteristic of tone [58]. The square function was processed in this study, but when considering the correlation, the mean pitch refers to the average frequency of the voice, with a higher mean pitch indicating a higher voice. The standard deviation pitch is calculated based on the average utterance pitch; the higher it is, the greater the change in spoken pitch. The ZCR is the rate by which the waveform crosses the horizontal line, which often performs as an index of whether voice is present in certain frames [59]. The voice portion refers to the ratio of the number of frames where the voice exists compared to the total amount of frames based on the ZCR.

According to the results of this study, in the order of ND, MDE, and mDE groups, the voice is lowered, and there are more pitch changes during speech. Even when the order of group 3 was confirmed through the Jonckheere–Terpstra test, it was statistically confirmed that the change in voice except for formant BW2 and sq_mean pitch was in the order of ND, MDE, and mDE (as shown in Table 3).

Previous research indicates that in depressed patients, the tone of voice becomes simpler, lifeless, and lower in volume [60,61]. Furthermore, a study comparing 47 depressed patients with 57 not depressed participants showed that the movement of the vocal tract was slow and participants spoke in low voices [28]. A further study comparing 36 depressed patients with a not depressed group also confirmed that depressed patients had low voices [22]. These results are consistent with the results of the present study, which also showed that the depressive group had lower voices than the control group.

Previous studies on the severity and pitch variability of depressed patients have shown contradictory results. In a 2004 study involving seven patients, a decrease in pitch variability was associated with depression severity; however, a 2007 study which analyzed 35 patients’ voices showed that the pitch variability increased as the depression increased [27,62]. In this study, it was confirmed that the change in voice pitch was greater in depressed participants than the control group. It was also confirmed that anxiety symptoms also increased with depression severity. Although further research is needed, anxiety can be expressed by the trembling of the voice and may possibly cause an increase in pitch variability.

Earlier research on the relationship between the severity of depression and voice characteristics was conducted on subjects diagnosed with depression. In this study, it was observed that the voice changes in the MDE were more pronounced than in the mDE. Minor depression, which has a higher prevalence than major depression, is considered a predecessor of and has a high likelihood of progressing to major depression [63]. However, minor depression is evaluated by including a group in which some of the symptoms have improved in major depression, i.e., a partial response [64]. In the present analysis, the mDE group was older than the MDE group, and 88.5% of the mDE group were taking antipsychotic medications at baseline. Considering the possibility that the mDE group had partially resolved depression, this suggests that even if the symptoms of depression improve, there is a possibility that the change in voice does not improve.

The result of predicting the severity of episodes using machine learning achieved 60.0% accuracy with an 8:2 train–test split in 93 cases. This accuracy of 60% is a reasonable level in three-group comparisons, and it is expected that the accuracy can be increased if the number of subjects is further increased. The reason why 60% accuracy in this study is acceptable is that in previous similar studies, the F1 score ranged from 0.303 to 0.633 depending on the system, and in natural language studies, the F1 score ranged from 0.51 to 0.71 [65,66]. Furthermore, previous studies made binary predictions to differentiate controls and depressed participants. However, in this analysis, since the case-wise inference was conducted to predict three groups with regard to the ND, mDE, and MDE, the accuracy was inevitably lower.

This study exhibits several strengths. Unlike previous studies, the subjects’ voices were recorded for a sufficient amount of time (mean 18 min) through semi-structured interviews. Thus, the audio files did not involve a mere repetition of sentences, but instead reflected various colloquial and paralinguistic expressions. Furthermore, unlike previous studies, statistical differences between vocal features and depression severity were confirmed, even after correcting clinical factors. Besides normalizing factors, such as age, use of other medicines, and BMI, the analysis was performed by considering the effects of antipsychotic medications on the voice and vocal cords. Importantly, the voice change in minor depression was also confirmed, which was found to be larger than in major depression. Therefore, this study highlights the potential for detecting and diagnosing minor depression through machine learning by using voice as a biomarker. It also suggests the possibility of using voice as an objective indicator when diagnosing major and minor depression. In addition, this study extracted 21 features of various voices; among the various indicators of voice, it was thought that there would be indicators that reflect the subject’s trait, such as gender, and there would be indicators that reflect the subject’s state. Therefore, since this study classified and analyzed many voice indicators, it may serve as a basis to inform the possibility of indicators reflecting state in voice.

This study has several limitations. As the first and most important limitation, the present study used a small sample size. Obtaining and pre-processing of the patient’s voice to extract its elements is a human resource-intensive activity. In this study, pre-processing was performed by marking both the start and end of the subject’s utterance while listening to the full interview, which took about three times the interview time. Thus, there have been obstacles in conducting such studies on a large scale. Therefore, it is necessary to develop a process that automatically discriminates the contents necessary for diagnosis based on the secured full interview recorded file. Furthermore, this study has the advantage that the average utterance time of each subject is long enough, but since the sample size is not sufficient, it is necessary to confirm whether the results of this study can be replicated through a larger-sized study. Secondly, although the clinical demographics were corrected and compared using ANCOVA, the voice indicators were not corrected by the degree of anxiety in each group. It is also possible that the degree of anxiety mediated the change of voice to a greater extent than the depression. In this regard, further research is needed, including mediation analysis of how the degree of anxiety changes the indicators of voice in depressed patients. Thirdly, this study was unable to confirm the relationship between the severity of depression and voice features in a cross-sectional way. Since the depressive symptoms improved while the voice changes did not, the potential effect of the drugs being taken cannot be excluded. Thus, future research should include an audio signal processing that automatically distinguishes the utterances of the interviewer from that of the subject. It may utilize recent methodologies that verify the speaker [67]. This can be augmented with conventional speech processing architecture to mitigate long-term temporal factors and multi-task inference.

## 5. Conclusions

This study reports preliminary indications that patients with depression exhibit lower voices and greater changes in pitch. Contrary to the hypothesis of this study, it was revealed that the voice function changed in the order of the ND, MDE, and mDE groups, respectively. However, the difference between the mDE and MDE groups was only observed in one of the 21 voices (standard deviation pitch). Further research in this area should include larger samples and follow-up studies on voice changes in minor depression.

## Figures and Tables

**Figure 1 jcm-10-03046-f001:**
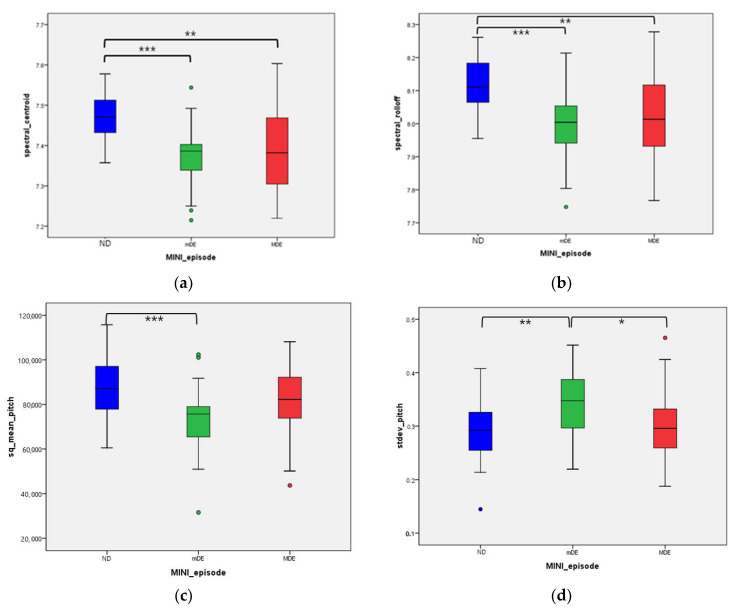
Difference of voice features by depressive episode by Benjamini–Hochberg test: (**a**) Spectral_centroid between three groups; (**b**) spectral_rolloff between three groups; (**c**) sq_mean_pitch between three groups; (**d**) stdev_pitch between three groups; (**e**) mean_magnitude between three groups; (**f**) zero-crossing-rate between three groups; (**g**) voice portion between three groups. * *p* value < 0.05, ** *p* value < 0.01, *** *p* value < 0.001. Abbreviations: ND—not depressed, mDE—minor depressive episode, MDE—major depressive episode, sqrt—square root, sq—squared, stdev—standard deviation.

**Figure 2 jcm-10-03046-f002:**
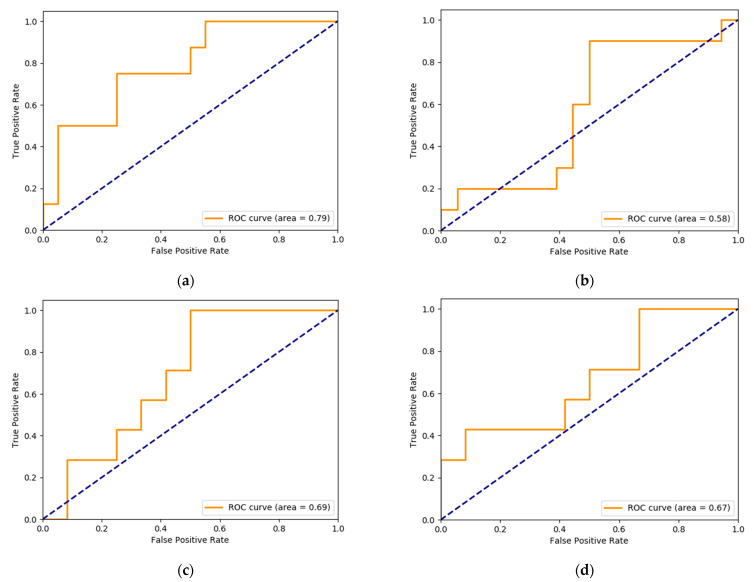
AUC curve predicting minor and major episodes using MLP: (**a**) AUC for minor episode, 7:3 training; (**b**) AUC for major episode, 7:3 training; (**c**) AUC for minor episode, 8:2 training; (**d**) AUC for major episode, 8:2 training; We only have the averaged result (for all the episodes) in Table 4, while this figure incorporates the result for each major and minor episode. Abbreviations: MLP—multi-layer perceptron, AUC—area under curve.

**Table 1 jcm-10-03046-t001:** Comparison of demographics according to depressive episodes.

		ND ^‡^	mDE	MDE	*p* Value	Post Hoc Test
N		33	26	34		
sex	M	8 (24.2%)	8 (30.8%)	7 (20.6%)	0.689	
	F	25 (75.8%)	18 (69.2%)	27 (79.4%)		
Age *		28.12 ± 4.827	34.58 ± 11.497	29.68 ± 9.914	0.022	
SES	Very low	0 (0%)	2 (7.7%)	1 (2.9%)	0.397	
	Low	10 (30.3%)	5 (19.2%)	7 (20.6%)		
	Middle	18 (54.5%)	15 (57.7%)	18 (52.9%)		
	High	5 (15.2%)	3 (11.5%)	4 (11.8%)		
	Very high	0 (0%)	1 (3.8%)	4 (11.8%)		
BMI ***		21.356 ± 1.861	23.470 ± 4.575	25.620 ± 5.396	<0.001	1 < 3 **
drugs taken for non-psychiatric disorders *	Yes	2 (6.1%)	8 (30.8%)	4 (11.8%)	0.025	1 ≠ 2 *
	No	31 (93.9%)	18 (69.2%)	30 (88.2%)		
Antipsychotics ***	Yes	0 (0%)	23 (88.5%)	29 (85.3%)	<0.001	1 ≠ 2 ***, 1 ≠ 3 ***
	No	33 (100%)	3 (11.5%)	5 (14.7%)		
Antipsychotics dose			5.142 ± 4.589	8.254 ± 7.893	0.100	
diagnosis by MINI ***	No psychiatric disorder	26 (78.8%)	0 (0%)	0 (0%)	0.000	
	Major depressive disorder	5 (15.2%)	9 (34.6%)	1 (2.9%)		
	Bipolar disorder	2 (6.1%)	17 (65.4%)	33 (97.1%)		
	Anxiety disorders ^†^	2 (6.1%)	2 (7.7%)	13 (38.2%)		
	Obsessive compulsive disorder	1 (3.0%)	1 (3.8%)	3 (8.8%)		
	Alcohol use disorder	2 (6.1%)	1 (3.8%)	3 (8.8%)		

* *p* value < 0.05, ** *p* value < 0.01, *** *p* value < 0.001. † Combined all types of anxiety disorders, including panic disorder, generalized anxiety disorder, and social anxiety disorder. ‡ Currently, there are no symptoms of depression, but past major episodes of depression are included. Abbreviations: ND—not depressed, mDE—minor depressive episode, MDE—major depressive episode, N—number, M—male, F—female, SES—social economic status, BMI—body mass index.

**Table 2 jcm-10-03046-t002:** Clinical characteristics by depressive episode.

		ND	mDE	MDE	*p* Value	Post Hoc Test
N		33	26	34		
HRDS ***	mean	3.879	13.346	18.706	<0.001	1 < 2 < 3 ***
	SD	2.902	4.127	4.414		
PHQ ***	mean	1.576	11.615	16.294	<0.001	1 < 2, 3 ***, 2 < 3 *
	SD	2.332	5.947	6.279		
BAI ***	mean	1.394	20.385	25.206	<0.001	1 < 2 ***, 1 < 3 ***
	SD	2.609	16.346	17.562		
BIS	mean	60.909	65.615	62.088	0.127	
	SD	5.598	8.750	8.155		

* *p* value < 0.05, ** *p* value < 0.01, *** *p* value < 0.001. Abbreviations: ND—not depressed, mDE—minor depressive episode, MDE—major depressive episode, N—number, HRDS—Hamilton depression rating scale, PHQ—patient health questionnaire, BAI—beck anxiety inventory, BIS—Barratt impulsivity scale, SD—standard deviation.

**Table 3 jcm-10-03046-t003:** Difference of voice features by depressive episode.

		ND	mDE	MDE	*p* Value	M–W Test	B–H Test	Adjusted *p* Value ^#^	J–T Test ^†^
N		33	26	34					
log_glottal_OP *	mean	0.890	0.846	0.925	0.037	2 < 3 *		0.051	0.381
	SD	0.105	0.096	0.137					
log_glottal_CP	mean	0.710	0.692	0.759	0.094			0.097	0.069
	SD	0.124	0.110	0.132					
log_glottal_C	mean	−0.675	−0.694	-0.627	0.094			0.097	0.068
	SD	0.124	0.110	0.132					
log_spectral_time	mean	2.345	2.583	2.457	0.191			0.461	0.343
	SD	0.463	0.529	0.497					
spectral_centroid ***	mean	7.471	7.375	7.398	<0.001	1 > 2 ***, 1 > 3 **	1 > 2 ***, 1 > 3 **	0.008 ^‡‡^	<0.001 ^†††^
	SD	0.058	0.075	0.105					
spectral_bandwidth	mean	7.444	7.422	7.430	0.343			0.968	0.384
	SD	0.050	0.057	0.069					
spectral_roll-off ***	mean	8.118	7.994	8.026	<0.001	1 > 2 ***, 1 > 3 **	1 > 2 ***, 1 > 3 **	0.012 ^‡^	0.001 ^††^
	SD	0.082	0.110	0.132					
spectral_rmse	mean	4.358	4.058	4.329	0.180			0.468	0.794
	SD	0.540	0.668	0.760					
log_spectral_tempo	mean	4.771	4.779	4.772	0.093			0.327	0.286
	SD	0.012	0.019	0.012					
formant1	mean	6.230	6.239	6.218	0.552			0.553	0.562
	SD	0.062	0.069	0.082					
formant2	mean	7.374	7.349	7.349	0.490			0.304	0.221
	SD	0.085	0.095	0.101					
formant3	mean	8.043	8.022	8.026	0.526			0.919	0.621
	SD	0.080	0.068	0.079					
formant_BW1	mean	42.083	39.681	44.119	0.331			0.159	0.729
	SD	10.233	8.399	14.107					
formant_BW2 *	mean	180.213	201.080	198.655	0.200			0.040 ^‡^	0.094
	SD	53.886	50.843	45.060					
sq_formant_BW3	mean	50622.267	52003.924	44566.728	0.133			0.094	0.292
	SD	13671.136	18982.013	14031.728					
sq_mean_pitch **	mean	87561.420	73835.557	81997.509	0.002	1 > 2 ***, 2 < 3 *	1 > 2 **,	0.027 ^‡^	0.149
	SD	12409.867	15014.241	15820.365					
stdev_pitch **	mean	0.287	0.344	0.300	0.003	1 < 2 **, 2 > 3 *	1 < 2 **, 2 > 3 *	0.020 ^‡^	0.520
	SD	0.057	0.065	0.067					
mean_magnitude **	mean	69.894	61.002	65.060	0.009	1 > 2**	1 > 2 *	0.059	0.110
	SD	11.454	11.292	9.902					
sq_stdev_magnitude	mean	0.787	0.748	0.852	0.140			0.237	
	SD	0.146	0.252	0.215					0.045 ^†^
ZCR ***	mean	0.055	0.044	0.047	<0.001	1 > 2 ***, 1 < 3 **	1 > 2 ***, 1 < 3 ***	<0.001 ^‡‡‡^	
	SD	0.007	0.006	0.010					< 0.001 ^†††^
voice portion **	mean	0.665	0.695	0.681	0.001	1 < 2 **, 1 < 3 *	1 < 2 **	0.020 ^‡^	
	SD	0.023	0.031	0.033					0.021 ^†^

* *p* value < 0.05, ** *p* value < 0.01, *** *p* value < 0.001. ^#^ Adjusted for BMI, age, non-psychiatric medication. ^‡^ adjusted *p* value < 0.05, ^‡‡^ adjusted *p* value < 0.01, ^‡‡‡^ adjusted *p* value < 0.001. ^†^ *p* value < 0.05, ^††^ *p* value < 0.01, ^†††^ *p* value < 0.001. Abbreviations: ND—not depressed, mDE—minor depressive episode, MDE—major depressive episode, M–W test—Mann–Whitney U test, B–H test—Benjamini–Hochberg test, J–T test—Jonckheere–Terpstra test, N—number, SD—standard deviation, OP—opening phase, CP—closing phase, C—closed phased, BW—bandwidth, ZCR—zero crossing rate.

**Table 4 jcm-10-03046-t004:** Machine learning model performance through voice features.

		7:3				8:2			
	Model	LR	GNB	SVM	MLP	LR	GNB	SVM	MLP
**Accuracy** **Mean (max)**	**augmented**	45.2 (53.6)	48.8 (57.1)	46.4 (50)	51.2 (53.6)	43.2 (57.9)	43.2 (57.9)	45.3 (52.6)	**60 (68.4)**
	**selected-augmented**	47.6 (57.1)	48.8 (57.1)	40.5 (53.6)	51.2 (57.1)	43.2 (63.2)	43.2 (57.9)	35.8 (42.1)	51.6 (57.9)
**AUC** **Mean (max)**	**augmented**	**63.4 (68)**	**64.7 (70.2)**	**60.3 (61)**	59.7 (65.1)	**64.5 (72.1)**	**64.5 (73.9)**	**61.1 (70.6)**	**65.9 (72.1)**
	**selected-augmented**	**63.6 (75.7)**	**63.6 (72)**	58.8 (66.5)	**62.9 (70.7)**	**62 (74.6)**	**60.3 (67.5)**	56.8 (64.9)	**62.6 (69.9)**
**Precision (sensitivity)** **Mean (max)**	**augmented**	45.1 (55.6)	49.1 (58)	47.3 (55)	51.4 (71.5)	46.6 (70.9)	41.6 (60)	48.7 (68.7)	**65.6 (76.7)**
	**selected-augmented**	57.2 (64.6)	54 (66.7)	36.3 (56.3)	**60 (62.6)**	44 (61.3)	42.3 (58.9)	34.6 (42.1)	**62.6 (72.2)**
**Recall (specificity)** **Mean (max)**	**augmented**	44.8 (55.1)	49.6 (60.2)	45.1 (48.6)	48.5 (55.8)	43.3 (61.7)	42.6 (64.2)	45.6 (52.5)	**66.2 (69.7)**
	**selected-augmented**	46.1 (55.8)	48.1 (56.7)	48.4 (64.6)	54.8 (62.5)	43.8 (64.6)	43 (58.1)	37.7 (44.2)	**93.3 (100)**
**F1** **Mean (max)**	**augmented**	43.1 (50.8)	46.4 (55.5)	43.5 (49.4)	44.3 (52.3)	42.1 (58.2)	39.6 (55.6)	43.4 (49.2)	58.9 (69.7)
	**selected-augmented**	45.4 (56.8)	47.8 (57.5)	40.7 (64.6)	49.3 (57.9)	41 (62.1)	42.1 (58.3)	33.1 (35.7)	58.7 (71.4)

Abbreviations: LR—logistic regression, GNB—Gaussian naïve Bayes, SVM—support vector machine, MLP—multi-layer perceptron, AUC—area under curve. Bold: mean or max is greater than or equal to 60.

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
