# Peer review of "Detection of Minor and Major Depression through Voice as a Biomarker Using Machine Learning"

_jcm, 2021, doi:10.3390/jcm10143046_

Round 1

Reviewer 1 Report

Thank you to the authors for addressing all concerns. I have no further comments.

Author Response

Thank you for taking your precious time to review.

Reviewer 2 Report

Thanks to the authors. My comments from before have been addressed for the most part. My comments below, oddly, are ordered from the least to most important

1. Based on the most recent change numbered 2., this should be 'drugs taken for non-psychiatric disorders" but this can be edited in final editing.

2. some other typos remain e.g. table 1 "disroder", "disorer"

3. Regarding the new emerging issue of mental illness in the control group:

  • Table 1 has a comment in the legend that "‡ Currently, there are no symptoms of depression, but includes past major episodes of depression." This should be, I suppose, in the ND group column and not in the row title cell of the table?
  • Should the above (i.e. current symptoms) be delineated also for the others - especially anxiety, bipolar and OCD but maybe also alcohol
  • This needs addressing further in discussion as it represents a significant barrier to interpreting results

4. I now notice that the vast majority (33/34) of the MDE sample met criteria for bipolar disorder - this is a huge element then of the study that may affect results and interpretation considerably. 97% of the MDE group vs 66% of the mDE group vs 6% ND. Were symptoms of current (hypo)mania measured? They are not discussed in either the methods, results or discussion - and are more than relevant in all. As well as these sections, part of the introduction (lines ~61) would need reworking too - one of the justifications for doing this research is that objective markers of depression are needed because antidepressants can cause hypomania but this therefore requires a differentiation of potential unipolar vs bipolar depression which your study could not inform - because bipolar depression is essentially being compared with a mixed sample of BD&UD who are not depressed.

Author Response

  1. Based on the most recent change numbered 2., this should be 'drugs taken for non-psychiatric disorders" but this can be edited in final editing.

 -> Your suggestion is correct. Table 1 was modified as suggested. (6P)

  1. some other typos remain e.g. table 1 "disroder", "disorer"

 -> There was a mistake and a typo in the table was corrected. In addition, the professional typo was checked again.

  1. Regarding the new emerging issue of mental illness in the control group:
  • Table 1 has a comment in the legend that "‡ Currently, there are no symptoms of depression, but includes past major episodes of depression." This should be, I suppose, in the ND group column and not in the row title cell of the table?
  • Should the above (i.e. current symptoms) be delineated also for the others - especially anxiety, bipolar and OCD but maybe also alcohol
  • This needs addressing further in discussion as it represents a significant barrier to interpreting results

-> As in your comment, it is important to consider things like compulsive and alcohol dependence symptoms together in the analysis.

 According to your suggestion, the legend has been changed to ND group.

 Although first diagnosed with bipolar disorder, in this study, subjective depression was required to participate in the study, and manic episodes were excluded. Therefore, being diagnosed with bipolar disorder at this time is unlikely to affect interpretation of the results.

 Also, although there are currently a small number of patients with anxiety disorders in the non-depression group, the mean of clinical anxiety symptoms as shown in Table 2 is significantly smaller than that in the mDE and MDE groups. It is judged that it is difficult to analyze the relationship between anxiety symptoms and voice with the current three-group classification. Therefore we plan to analyze the mediation of anxiety symptoms on the effects of depressive symptoms on voice in the future. These current analysis limitations have already been described in the discussion. (15P, line 389-393)

 In addition, some patients with OCD and alcohol use disorder currently have symptoms, but they were not included as covariates because the difference in the number of patients in each group was not significant. If such a study is conducted on a large scale in the future, it seems that various clinical symptoms should be fully considered as you suggested.

  1. I now notice that the vast majority (33/34) of the MDE sample met criteria for bipolar disorder - this is a huge element then of the study that may affect results and interpretation considerably. 97% of the MDE group vs 66% of the mDE group vs 6% ND. Were symptoms of current (hypo)mania measured? They are not discussed in either the methods, results or discussion - and are more than relevant in all. As well as these sections, part of the introduction (lines ~61) would need reworking too - one of the justifications for doing this research is that objective markers of depression are needed because antidepressants can cause hypomania but this therefore requires a differentiation of potential unipolar vs bipolar depression which your study could not inform - because bipolar depression is essentially being compared with a mixed sample of BD&UD who are not depressed.

 -> You pointed out a important point. Thanks for telling us about this. We did not evaluate hypomanic symptoms because we considered depressive conditions from the initial design of this study. When this study is further expanded in the future, it seems to be very important to design so that the manic symptoms can be evaluated together with your advice. Thanks again for your advice.

Reviewer 3 Report

Very interesting paper, very well presented and clear results and methodology. It's a novelty in the way of detecting and treating depression. 

The question in objective of this paper is if voice could be used as a biomarker to detect minor and major depression, which is an interesting approach in detecting depression in addition to all the questionnaires available in today's diagnosis system.

I found this paper original since I always see questionnaires as a principal method in the diagnosis of depression amongst patients. The paper is well written and very explanatory.

The text is clear and easy to read and the conclusions are consistent with all the evidence and arguments presented which are many with the voice test made for the patients.

They address the main question with introductory evidence and clinical evidence with the voice tests made with the sample that participated in the study.

Author Response

Thank you for taking your valuable time to review.
We also use questionnaires to diagnose patients.
This study tried to overcome the limitations of the questionnaire and investigate the possibility of a new diagnosis.
Thank you again for understanding and acknowledging the value of this research.

This manuscript is a resubmission of an earlier submission. The following is a list of the peer review reports and author responses from that submission.

Round 1

Reviewer 1 Report

Thank you to the authors for responding to the reviewers’ comments. A few comments and concerns remain.

  1. In regards to defining depression.
    1. It would be important to add that in order to be diagnosed with major depression, at least one of the symptoms should be either (1) depressed mood or (2) loss of interest or pleasure.
    2. There is a sentence that reads: “However, since 1992, the importance of depression, which is not suitable for major depression, has a short duration of depression or a mild symptom of an insufficient number of depression symptoms has been revealed.”
      1. This is a bit confusing. Are the authors referring to mild depression? Is it also based on the DSM-V? The DSM-V criteria for minor depressive episodes or minor depression is when two to four of the symptoms have been present during the same two-week period.
    3. The DSM-V is already spelled in the 1st Thus, the authors can use DSM-V in the 2nd paragraph.
  2. Voices as standard by which clinicians judge patient symptoms
    1. I believe that other clinicians will also wonder what is the difference between voice and speech, as we often focus on speech when we evaluate patients. As such, it would be worth clarifying in the introduction that you measured both in this study, and even incorporating in the discussion.
  3. MINI
    1. There are different versions. I believe the most recent is the 7.0.2
  4. Impulsiveness
    1. While I understand the authors’ argument, it continues to be my opinion that this variable distracts from the main aims of the study. I suggest that the authors reconsider whether it is important to keep this variable in the paper.

Author Response

  1. In regards to defining depression.

1-1 It would be important to add that in order to be diagnosed with major depression, at least one of the symptoms should be either (1) depressed mood or (2) loss of interest or pleasure.

 à The two diagnostic criteria you mentioned are an important part. Therefore, I added the following to the introduction.

 “Based on the Diagnostic and Statistical Manual of Mental Diseases, Fifth Edition (DSM-V), a major depressive disorder can be diagnosed when 5 or more different depressive symptoms, including one or more (1) depressed mood or (2) loss of interest or pleasure, lasting more than 2 weeks or longer”

1-2 There is a sentence that reads: “However, since 1992, the importance of depression, which is not suitable for major depression, has a short duration of depression or a mild symptom of an insufficient number of depression symptoms has been revealed.”

 à Since the meaning of the sentence is not clear, it has been modified as follows.

 “However, the importance of minor depression has been revealed since 1992. Minor depression was diagnosed as unsuitable for major depression, such as a short period of depression, not being satisfied with either depression or decreased interest, or having only 4 or fewer depressive symptoms”

1-3 This is a bit confusing. Are the authors referring to mild depression? Is it also based on the DSM-V? The DSM-V criteria for minor depressive episodes or minor depression is when two to four of the symptoms have been present during the same two-week period.

 à As stated in lines 108~113, we did not define minor depression based on DSM-5, and the subjects who did not satisfy the episode in MINI but expressed subjective depression were defined as minor depression. For this reason, the group was named as a minor depressive episode group.

1-4 The DSM-V is already spelled in the 1st Thus, the authors can use DSM-V in the 2nd paragraph.

 à Thank you for telling us. From the second paragraph, it was used as DSM-V.

  1. Voices as standard by which clinicians judge patient symptoms

I believe that other clinicians will also wonder what is the difference between voice and speech, as we often focus on speech when we evaluate patients. As such, it would be worth clarifying in the introduction that you measured both in this study, and even incorporating in the discussion.

 à Thank you for your comment. We have also added that content to the introduction.

“Typically, voice is an index that reflects the characteristics of the vocal cords, and speech is an index that includes speech speed and hesitation” in line 66~67

 We also added about voice and speech in discussion part.

“Among the 21 various indicators, various indicators are included, such as the average of the pitch reflecting the characteristics of the voice and the ratio of the actual utterance to the utterance time reflecting the delay of speech.” In lint 299~302.

  1. MINI

There are different versions. I believe the most recent is the 7.0.2

 à The version we used was 7.0.2. The version was also added in line 127, method part.

  1. Impulsiveness

While I understand the authors’ argument, it continues to be my opinion that this variable distracts from the main aims of the study. I suggest that the authors reconsider whether it is important to keep this variable in the paper.

 à We thought about whether it would be better to delete the content on impulsiveness. However, we thought that it may not be ethical to exclude from paper because the results measured in the study were not statistically significant.

Reviewer 2 Report

I believe that overall, this version is improved. Most of the below comments are minor, but the latter numbered points below are progressively more important.

  1. Abstract - the change to wording (now "dimensional dimension") doesn't make sense, I would suggest dimensional 'categorisation' or 'construct'
  2. Abstract - "other drug use" should be phrased more specifically e.g. "non-psychotropic"?
  3. It would be useful to add the sample size limitation explicitly to the end of the abstract
  4. Many of the revisions require some editing for grammatical English 
  5. Table 3 - this requires formatting (currently hard to read) and the row headings need to be rewritten as words / constructs (at the moment they are what looks like variables from a database)
  6. Figure 1 - Axis headings on the figure should be worded descriptively rather than using unclear variable names
  7. Discussion - lines 403-404 - I'm not sure that (as the authors state) their findings are really consistent with the study mentioned above that. It looks to me that the minor-depression group's voices were lower, but the major-depression group had higher voices than the minor group and not statistically different from the ND group? This is also the case with some of the other discussion of findings, to my understanding. Please could the authors check and clarify in the discussion.
  8. Discussion - lines 423-424 - i'm not sure that even with a three group comparison AUC 60 is good, but I couldn't easily check this because no reference was given next to this statement. Please could one be added?
  9. Discussion (general) - it doesn't come across to me that sufficient emphasis has been placed on the sample size limitation with these results. 
  10. Discussion (conclusion) - line 494 - related to above, request for the authors to reword "confirmed" to something like "reports preliminary indications that", because there is not statistical power to really confirm anything with this sample size.

Author Response

  1. Abstract - the change to wording (now "dimensional dimension") doesn't make sense, I would suggest dimensional 'categorisation' or 'construct'

 à The word could not be deleted during the correction process. It was modified to “based on current depressive status as a dimension”.

  1. Abstract - "other drug use" should be phrased more specifically e.g. "non-psychotropic"?

 à Like your comment, it seems that the meaning of the word is not clear. Therefore, it has been modified as follows. “drugs taking for non-psychiatric disorders”

  1. It would be useful to add the sample size limitation explicitly to the end of the abstract

 à I agree with your comment, and added the following to the abstract. “Although this study is limited by insufficient sample size,”

  1. Many of the revisions require some editing for grammatical English

 à Thank you for your comment. We received additional English corrections for the revised part.

  1. Table 3 - this requires formatting (currently hard to read) and the row headings need to be rewritten as words / constructs (at the moment they are what looks like variables from a database)

 à We tried to fix it as neatly as possible on that part.

  1. Figure 1 - Axis headings on the figure should be worded descriptively rather than using unclear variable names

 à We agree with your opinion. However, it is difficult to express the descriptive meaning of each variable in one word. Therefore, we wrote about what each feature means in the discussion. Also, it seems that it will help readers to find the variables that the negative variables are written with the same names as the variables in Table 3.

  1. Discussion - lines 403-404 - I'm not sure that (as the authors state) their findings are really consistent with the study mentioned above that. It looks to me that the minor-depression group's voices were lower, but the major-depression group had higher voices than the minor group and not statistically different from the ND group? This is also the case with some of the other discussion of findings, to my understanding. Please could the authors check and clarify in the discussion.

à Both variables spectral_centroid and spectral_roll-off have statistical significance between ND and mDE and between ND and MDE. Also, there was no statistical significance only between mDE and MDE. This is also represented in table 3 and also in figure 1. Therefore, it was described that voices lower in both major and minor depression episode groups than ND.

  1. Discussion - lines 423-424 - i'm not sure that even with a three group comparison AUC 60 is good, but I couldn't easily check this because no reference was given next to this statement. Please could one be added?

à In the next sentence of that sentence, the rationale for why accuracy is reasonable is explained. “The accuracy of 60% is a reasonable level of accuracy in three-group comparison, and it is expected that the accuracy can be increased if the number of subjects is further increased. The reason why the 60% accuracy in this study is reasonable is that in previous similar studies, the F1 score ranged from 0.303 to 0.633 depending on the system, and in natural language studies, the F1 score ranged from 0.51 to 0.71 [65, 66].”

  1. Discussion (general) - it doesn't come across to me that sufficient emphasis has been placed on the sample size limitation with these results.

 à  We are very sorry for being considered that way. However, as your advice, the abstract and conclusion have sufficiently mentioned the sample size as a limitation, and the discussion also presented the limitation for the first time. This seems to be insufficient as well, so I modified it to “As the first and most important limitation as the first mention of the limitation.

Although this study did not have a larger sample size than the previous studies, but it has a sample size that is not smaller than that of the previous studies. And this study has the advantage that the utterance time obtained from one subject is longer.

With the help of your advice, it seems that the sample size has been developed to fully address the limitations of this study, so please consider positively.

  1. Discussion (conclusion) - line 494 - related to above, request for the authors to reword "confirmed" to something like "reports preliminary indications that", because there is not statistical power to really confirm anything with this sample size.

 à We agreed with your opinion and revised it as the conclusion suggested.